# Occurrence of Surgical Site Infections at a Tertiary Healthcare Facility in Abuja, Nigeria: A Prospective Observational Study

**DOI:** 10.3390/medsci6030060

**Published:** 2018-07-30

**Authors:** Ahmed Olowo-okere, Yakubu Kokori Enevene Ibrahim, Ali Samuel Sani, Busayo Olalekan Olayinka

**Affiliations:** 1Department of Pharmaceutics and Pharmaceutical Microbiology, Usmanu Danfodiyo University, Sokoto 840, Nigeria; 2Department of Pharmaceutics and Pharmaceutical Microbiology, Ahmadu Bello University, Zaria 810, Nigeria; ykeibrahim@gmail.com (Y.K.E.I.); busayoolayinka@yahoo.com (B.O.O.); 3Department of Surgery, University of Abuja Teaching Hospital, Gwagwalada, FCT-Abuja 902, Nigeria; sambecca_2002@yahoo.com

**Keywords:** healthcare associated infections, surgical wound infections, SSI, risk factors, epidemiology

## Abstract

Surgical site infection (SSI) is one of the most frequent complications of surgical interventions. Several factors have been identified as major determinants of occurrence of SSIs. The present study determined the occurrence and possible risk factors associated with SSIs at a tertiary healthcare facility in Abuja, Nigeria. All patients scheduled for operation in the hospital during the study period and who consented to participate willingly in the study were observed prospectively for the occurrence of SSI based on criteria stipulated by the United States Centre for Disease Control and Prevention (CDC). Data on sociodemographic characteristics, lifestyle, surgical procedure and co-morbidity were collected into a pre-tested data collection tool and analysed using IBM SPSS Statistics software v.24. Predictors of SSIs were identified using multivariate logistic regression model and *p*-value less than 0.05 was considered statistically significant. Of the 127 surgical patients that met the inclusion criteria comprising 65 (51.2%) females and 62 (48.8%) males between 1 and 83 years with mean age of 25.64 ± 1.66 years, 35 (27.56%; 95% Confidence Interval (CI): 0.205–0.360) developed SSIs. Prolonged post-operative hospital stays (*p* < 0.05), class of wound (*p* < 0.0001) and some comorbid conditions were found to be significantly associated with higher SSI rate. The SSI rate was highest among patients that had Kirschner-wire insertion (75.0%), followed by an unexpectedly high infection rate among patients that had mastectomy (42.9%), while lower percentages (33.3%) were recorded among patients that had exploratory laparotomy and appendicectomy. The overall magnitude of SSIs in this facility is high (27.6%; 95% CI: 0.205–0.360). Several factors were found to be independent predictors of occurrence of SSI. The findings thus highlight the need for improved surveillance of SSIs and review of infection control policies of the hospital.

## 1. Introduction

Whilst surgical site infection (SSI) has been defined in various ways, the most widely used and accepted definition is the 1992 reclassification and definition by United States of America Centre for Disease Control and Prevention (CDC) which defined SSI as “an infection that occurs after surgery in the part of the body where the surgery took place within 30 days of an operative procedure or within one year if an implant is left in place” [1,2].

SSIs increase the length of postoperative hospital stay, escalate the cost of healthcare, increase the rate of hospital readmissions, and jeopardize health outcomes [3]. It is sometimes associated with considerable morbidity and occasional mortality [1]. Approximately ten additional days on the length of hospital stay have been attributed to SSIs [4]. The healthcare cost for patients with SSIs has been estimated to be twice that of the patients without SSI [4,5]. In addition, increased relative risks of death and readmission for patients with SSI compared to uninfected patients have been documented [6].

Studies involving analysis of risk factors have shown that most SSIs are attributable to patient-related factors rather than procedure-related factors [7,8]. These factors directly suppress the patient’s immunity and patient’s ability to recover from surgical incisions [9]. These factors decrease neutrophil bactericidal activity thereby making the body’s natural defence system impotent in the face of microbial contaminants [10]. The factors include diabetes mellitus, smoking, alcoholism, cancer, chronic renal failure, jaundice, obesity, advanced age, poor physical condition, medications such as steroids and antineoplastic drugs and previous radiotherapy or chemotherapy, malnutrition, pre-existing infections, preoperative hospital stay, immunodeficiency and colonization with *Staphylococcus aureus* or other potential pathogens [7,8,10].

The occurrence of post-operative wound infections varies widely between procedures, hospitals, surgeons, patients and geographical locations [8,11]. In developing countries, lack of expertise and resources required for effective surveillance of hospital acquired infections limit availability of data on its occurrences [12,13]. As such, the magnitude is either underestimated or largely unknown. At the largest National hospital in Tanzania, 26% SSI incidence rate has been reported [11]. In another study on the incidence of SSI among patients that had caesarean section (CS) in Tanzania, an overall incidence of 10.9% SSI was reported [14]. In the same study, the median time from CS to the development of SSI was seven days and six independent risk factors for post caesarean SSI were identified. A study conducted at a rural setting in Gabon reported a low incidence of 1.6% SSI [15].

While the incidence of SSIs in many parts of Nigeria has been documented, there are no data on its status in Abuja and its environ. The present study therefore aimed to determine the occurrence and possible risk factors associated with SSIs at a tertiary healthcare facility in Abuja, Nigeria.

## 2. Methods

### 2.1. Description of Study Centre

This study was conducted at the surgical wards (adult and children) as well as operating theatres of University of Abuja Teaching Hospital (UATH), Gwagwalada. The hospital is a tertiary hospital providing in- and outpatient medical services to residents of Federal Capital Territory and nearby towns and villages in adjoining states. It has 350 bed spaces with facility to expand to 500 beds. The hospital has eight well equipped operating suites with 13 consultant surgeons in different surgical specialties, 12 senior registrars and a good number of resident doctors and house officers. The hospital has four surgical wards (male, female, paediatric and obstetric and gynaecological wards). The wards have 32 beds each.

### 2.2. Determination of Sample Size and Sampling Method

A simple random sampling technique was used and the sample size was determined using the Thrushfield (2005) formula [16]:(1)$$\mathrm{Sample}\;\mathrm{size}\;\left(n\right)=z^{2}\times \frac{Ρ\left(1-Ρ\right)}{d^{2}}$$

The sample size (*n*) was determined based on 15.5% [17] expected prevalence rate (P), absolute desired precision (d) of 5% at confidence interval (CI) of 95% and z^2^ (standard normal variate) which at 5% Type 1 error is 1.96.
*n* = 1.96^2^ × 0.155(1−0.155)/0.05^2^ = 201.26 ≈ 201 patients

### 2.3. Study Design and Patient Selection Criteria

This was a prospective cohort study involving all patients scheduled for surgical operations in the hospital that willingly consented to participate in this study with the exception of patients that were admitted with infected post-operative wound requiring re-operation, obstetrics and gynaecological patients, patients with infected burn wounds and patients that had day cases operations such as tonsillectomy, excision of lumps, etc as well as patients that developed SSI after discharge from the hospital.

### 2.4. Monitoring Patients for Development of Surgical Site Infections

All patients scheduled for operation in the hospital, between 1 March and 31 May 2015, that met the inclusion criteria and consented to willingly participate in the study were recruited. Data on socio-demographic and other patients related factors were extracted directly from patients’ folder and patients’ operation note into pre-tested data collection tool. The data collected include age in years, gender, admission and discharge date, date of operation, type and nature of surgery, wound class (clean or clean-contaminated), and co-morbidity status of the patients (diabetes mellitus, pre-existing infection and anaemia). Information on smoking habit and alcohol consumption were also collected.

All operated patients received prophylactic and post-operative antibiotic covering, for 5–7 days. The most commonly used antibiotics is the combination of a third-generation cephalosporin (ceftriaxone) and either metronidazole or ciprofloxacin.

During the period of convalescence in the hospital wards, patients were observed by the attending surgeons for the development of SSIs by careful and meticulous examination of the operated site on the day of first wound dressing and subsequently before each wound dressing until the patients were discharged from the hospital, 30 days post-operative for patients on admission for more than 30 days and up to 90 days for patients with implants [2].

The diagnosis of operated sites as positive for SSIs and subsequent categorization into superficial incisional, deep incisional or organ/space SSIs was based on the criteria stipulated by the CDC guideline [2].

### 2.5. Data Analysis

Data collected were analysed using IBM SPSS statistics for windows, version 24 (IBM Corp., Armonk, NY, USA). Descriptive statistics (frequency tables and cross tabulations) were used in the analysis. Categorical variables were expressed in percentages and chi-square test was used to determine the association between independent and dependent variables. Multivariate logistic regression model was used to identify the predictors of SSIs. The finding was presented using adjusted odds ratio (OR) and their 95% confidence interval. All *p*-values were two-sided, and values less than 0.05 were considered statistically significant.

### 2.6. Ethical Consideration and Recruitment of Patients for the Study

The approval to conduct this study was given by Health Research Ethics Committee of the hospital (FCT/UATH/HREC/PR/389). After obtaining ethical approval, in line with World Health Organisation (WHO) requirement on biomedical research, written informed consent of each patient or their guardians to willingly participate in the study was sought. Patients confidentiality was maintained throughout the study period and the participants were adequately informed of their right to withdraw from the research if they so wish without any implication.

## 3. Results

### 3.1. Incidence of Surgical Site Infections

Out of a total of 135 patients admitted during the study period, 127 patients, comprising 65 (51.2%) females and 62 (48.8%) males, met the inclusion criteria. The age of the patients ranged between 1 and 83 years with mean age of 25.6 ± 1.66 years. Of these patients, 35 (27.6%; CI: 0.205–0.360) developed SSIs

### 3.2. Patients Related Factors

The distribution of the observed occurrence of SSIs among the various age groups and gender of patients enrolled for the study revealed that infection rates of 29.2% and 25.8% were observed among the female and male patients, respectively (Table 1). Among the various age groups of patients observed, patients aged 50 years and above had the highest rate (35.3%) while patients aged 19–49 years have the lowest rate (23%).

Thirty-six of the patients had one or more co-morbidities namely diabetic mellitus, sickle cell disease, anaemia and pre-existing infection. Two (33.3%), three (37.5%) and five (31.3%) of the patients with diabetes mellitus, anaemia and pre-existing infection, respectively, had SSIs (Table 1).

Analysis of the infection rates using chi square test of independence shows that there was no significant difference in infection rates among the genders (*p* = 0.666), age categories (*p* = 0.500), types of surgery (*p* = 0.233) and alcohol consumption (*p* = 0.189). However, highly significant statistically relationship was found between duration of post-operative hospital stay (*p* < 0.01) and class of wound (*p* < 0.0001) and the occurrence of SSI. The mean duration of post-operative hospital stay was 21.94 ± 2.45 for patients with SSIs and 14.81 ± 1.08 for patients without SSIs. The patients with SSIs spent an additional 7.1 days on average in the hospital compared to their counterpart without SSIs (mean difference, 7.13; 95% CI: 1.75–12.52; *p* < 0.05).

### 3.3. Procedure Related Factors

Among the various types of operation observed, infection rate was highest among patients that had Kirschner-wire insertion (75.0%). This was distantly followed by an unexpected high infection rate among patients that had mastectomy (42.9%), while lower percentages were recorded among patients that had exploratory laparotomy (33.3%) and appendicectomy (32.3%) (Figure 1).

Dirty wounds were the most infected (66.67%) wound class while the lowest rate was observed in the clean wounds (Table 1). Chi-square test of independence revealed very highly significant association between the wound class and the development of SSIs (*p* < 0.001). The superficial incisional site (22; 62.9%) was the most frequently infected surgical site, followed by deep incisional sites (9; 25.7%) and organ/space (4; 11.4%) infection sites (Figure 2).

### 3.4. Predictors of Surgical Site Infection

Multivariate logistic regression was performed to assess the impact of a number of risk factors on the likelihood of occurrence of SSI. As shown in Table 2, advancing age (adjusted odd ratio = 1.02; 95% CI: 0.993–1.055), post-operative hospital stays (OR = 1.07; 95% CI: 1.011–1.131), diabetes mellitus (OR = 1.2; CI: 0159–9.109) and cigarette smoking (OR = 6.24; 95% CI: 0.274–142.15) showed higher likelihood of occurrence of SSI compared to their counterparts. However, only prolonged post-operative hospital stays (*p* = 0.02) significantly contributed to the model.

## 4. Discussion

The result of this study revealed a high incidence rate of SSIs (27.6%). In the same centre, we reported 15.6% and 13.6% SSI prevalence rates in 2013 and 2014 studies, respectively [18]. This is comparable to high incidences of SSIs reported across Nigeria [19,20,21,22,23], Ethiopia (19.1%) [24] and Tanzania (26.0%) [25]. It was however, higher than figures reported in Egypt [26], Ghana [27], England [28] and USA [29]. The SSI incidence rates obtained from various centres across the country and internationally may not be comparable as surveillance criteria, patients’ characteristics, wound classifications and hospital policy on the use of invasive vs. open procedures may differ. In addition, criteria for inclusion and exclusion of patients for the study may also differ. Thus, there is need for standardization and institutionalization of surveillance system in the country to permit valid comparison of surveillance results. Differences in the availability of state-of-the-art healthcare infrastructure, skilled and motived manpower and application of standard infection control practices may be responsible for the discrepancies in the incidence of SSIs reported in the present study compared to the high-income countries.

The higher infection rate seen among patients aged ≥50 years or ≤18 years is consistent with findings of other researchers [30]. This might be because age is an important patient-related factor that can predispose patients to post-surgical wound infections [7]. Underdeveloped organ system in children and debilitating organs in the elderly cannot mount effective response to infectious challenges and thus higher infection rate is always expected in these age categories and as such, extra caution is always required [31].

Consistent with the reports of other researchers, dirty and contaminated wound classes were the most infected [17,32,33]. Contamination of operated site has been reported to be the single most important factor determining the occurrence of SSI [10]. In contaminated and dirty wounds, contaminating pathogens were already present at the operating sites prior to commencement of the surgery [1]. Pre- and post-operative antibiotics therapies would have been expected to significantly reduce this threat to a level that can be effectively eliminated (inhibited or killed) by the activated host immune system [10]. However, immune system of most hospitalized patients is compromised from malnutrition during hospitalization, disease condition such as HIV/AIDS or drugs such as cytotoxic and steroids [9]. In addition, harbouring of resistant genes with possibility of horizontal transfer of these resistant genes coupled with intrinsic antimicrobial resistance in some bacteria may explain why some of the bacteria survived competent host immune response and antibiotics to cause clinical infections [34,35]. According to Owens and Stoessel (2008), antibiotic-resistant pathogens such as methicillin-resistant *Staphylococcus aureus*, vancomycin-resistant enterococci, and extended spectrum beta-lactamase-producing Enterobacteriaceae are responsible for significant number of SSI [7]. Additionally, various materials used in the care of surgical patients such as surgical sutures, intravascular catheters, draining systems, prosthetic devices, etc. may provide a point of anchor for bacteria which produce protective extracellular matrix containing polysaccharide, DNA and proteins leading to biofilm formation [36]. It has been estimated that 80% of SSI may involve biofilm, which through its organized structural arrangement protects bacteria from both intrinsic (host immune system) and extrinsic (antimicrobials) threats [37]. Biofilm has been demonstrated to contribute to recalcitrance SSI which may persist for months [36,38]. In Nigeria, role of biofilms in SSI development has not been sufficiently investigated. This is an important area of future research.

The pattern of occurrence of SSI among the various types of operation in this study is consistent with the findings of another researcher [39]. Most of the patients that had mastectomy in this study had an established clinical infection before the surgery with breast abscess in some cases. This may explain the observed high infection rate in operations that were otherwise clean.

The patients with SSIs are more likely to spend additional days in the hospital compared to their counterpart without SSIs. This is consistent with several other reports [4,12]. The higher occurrence of SSI among patient with prolonged postoperative hospital stay may be due to prolong exposure of the patients to nosocomial pathogens coupled with weakened immunity occasioned by prolong hospitalization [9].

The findings of higher likelihood of occurrence of SSI among patients with advanced age (OR = 1.02; 95% CI: 0.993–1.055), prolonged post-operative hospital stays (OR = 1.07; 95% CI: 1.011–1.131), diabetes mellitus (OR = 1.2; CI: 0159–9.109) and cigarette smoking (OR = 6.24; 95% CI: 0.274–142.15) compared to their counterparts supports the reports of other studies where these factors have been identified as independent predictors of SSIs [24]. Patients who smoke cigarette have almost six times the risk of SSIs than their counterparts who do not smoke cigarettes (OR: 6.24; 95% CI: 0.274–142.15). This concurs with the finding of Nwankwo (2012) [33]. Because of small number of patients with various co-morbid conditions during the study period, statistically significant relationship could not be established. Other researchers have found these factors to significantly predispose patients to SSIs [10,32,40,41,42].

This study was intermittently interrupted by industrial actions embarked upon by Nigeria Medical Association (NMA), Joint Health Sector Union (JOHESU) and Association of Resident Doctors (ARD). As a result, the targeted number of cases could not be investigated. A multidisciplinary research involving all the surgical service units in the hospital over an extended period would give a better picture of SSIs incidence rate in the hospital. This study however provides baseline data on the occurrence of SSIs at the hospital and it may serve as a useful tool in decision making and in allocating limited resources and expertise in addressing SSI and nosocomial infections in general in the hospital.

Surveillance studies on the incidence of SSIs across the country should be standardized and if possible automated to increase its efficiency. In addition, post discharge surveillance method should be developed as a good number of SSIs have been detected after discharge from hospital.

Multidisciplinary Nosocomial Infection Surveillance Committee should be set up and equipped to respond adequately to the trend of nosocomial infection in the hospital.

The level of awareness of nosocomial SSIs among the surgeons and other health care personnel in the hospital though commendable should be improved through awareness campaigns geared towards sensitizing the personnel on the risk factors associated with SSIs; its impacts on the patients, health care facilities and resources; and preventive measures.

## 5. Conclusions

The overall magnitude of SSIs in this facility is high (27.6%; 95% CI: 0.205–0.360). Several factors such as the duration of post-operative hospital admission, class of wounds and present of some co-morbid conditions were found to be independent predictors of occurrence of SSI. The findings highlight the need for improved surveillance of SSIs and review of infection control policies of the hospital.

## Figures and Tables

**Figure 1 medsci-06-00060-f001:**
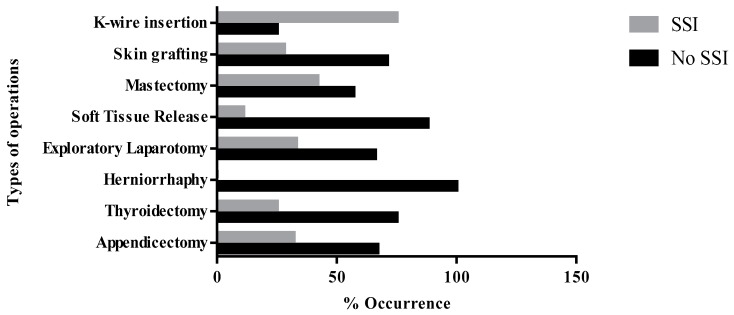
Type of operations and occurrence of SSI.

**Figure 2 medsci-06-00060-f002:**
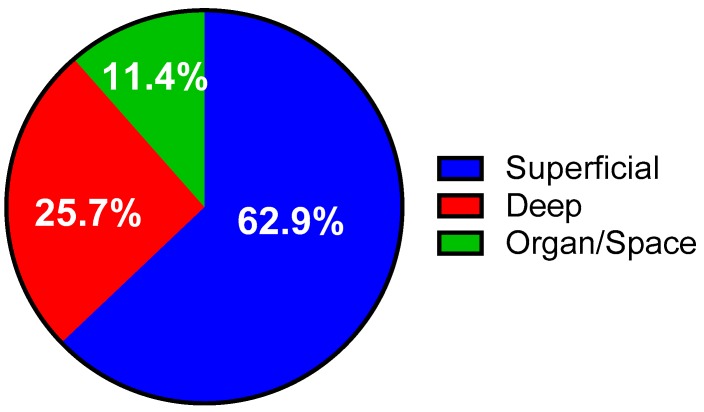
Types of SSI.

**Table 1 medsci-06-00060-t001:** Relationship between various patients and clinical parameters and incidence of surgical site infection (SSI).

Parameters	SSI Present Frequency (%)	SSI Absent Frequency (%)	Χ^2^	*p*-Value
**Gender**				
Female	19 (29.2)	46 (70.8)	0.186	0.666
Male	16 (25.8)	46 (74.2)		
**Age Categories**				
0–18 Years	15 (30.6)	34 (69.4)	1.387	0.500
19–50 Years	14 (23.0)	47 (77.0)		
>50 Years	6 (35.3)	11 (64.7)		
**Types of Surgery**				
Elective	17 (33.3)	34 (66.7)	1.423	0.233
Emergency	18 (23.7)	58 (76.3)		
**Nature of Wound**				
Clean	3 (6.5)	43 (93.5)	23.1	<0.0001
Clean-Contaminated	11 (30.6)	25 (69.4)		
Contaminated	11 (36.7)	19 (63.3)		
Dirty	10 (66.7)	5 (33.3)		
**Co-Morbid Factor**				
*Alcohol Consumption*				
Yes	2 (13.33)	13 (86.67)	1.724	0.189
No	33 (29.46)	79 (70.54)		
*Cigarette Smoking*				
Yes	2 (50.0)	2 (50.0)	1.042	0.307
No	33 (26.8)	90 (73.2)		
*Pre-Existing Infection*				
Yes	5 (31.25%)	11 (68.75%)	0.125	0.724
No	30 (27.03%)	81 (72.97%)		
*Anaemia*				
Yes	3 (37.50%)	5 (62.50%)	0.4230	0.516
No	32 (26.89%)	87 (73.11%)		
*Diabetes mellitus*				
Yes	2 (33.33%)	4 (66.67%)	0.105	0.746
No	33 (27.27%)	88 (72.73%)		

**Table 2 medsci-06-00060-t002:** Multivariate logistic model on predictors of SSI at surgical ward of the hospital.

Variables	B	S.E.	Wald	Sig.	Exp (B)	95% CI
Age	0.023	0.015	2.344	0.126	1.024	0.993–1.055
Post-operative hospital stays	0.067	0.029	5.402	0.020	1.069	1.011–1.131
Nature of operation (Emergency)	−1.037	0.601	2.970	0.085	0.355	0.109–1.153
Diabetes mellitus	0.186	1.032	0.033	0.857	1.205	0.159–9.109
Anaemia	−0.045	1.022	0.002	0.965	0.956	0.129–7.084
Pre-existing Infection	−0.186	0.811	0.053	0.818	0.830	0.169–4.071
Cigarette smoking	1.830	1.595	1.316	0.251	6.236	0.274–142.154
Alcohol Consumption	−1.666	1.228	1.843	0.175	0.189	0.017–2.095
Constant	−1.216	0.731	2.762	0.097	0.297	

B: coefficient; S.E: standard error; Wald: Wald test; Sig: *p*-value; Exp(B): odds ratio; CI: confidence interval.

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
