# Peer review of "Occurrence of Surgical Site Infections at a Tertiary Healthcare Facility in Abuja, Nigeria: A Prospective Observational Study"

_medsci, 2018, doi:10.3390/medsci6030060_

Round 1

Reviewer 1 Report

Comments

Introduction- 

1.    Rewrite first few lines of introduction keeping only the CDC defination.

2.    Page 2 line 49-50 are factually incorrect, please rewrite. Eg of surgery related factors are-duration of surgery, experience of surgion, hypoxia during surgery, surgical prophylaxis.

Methods-The “surgical prophylaxis” of 5 days cannot be called prophylaxis according to CDC defination

3.    What was the defination of SSI used for the study? 

4.    What criteria were met to define SSI? 

5.    It is unclear if cultures were sent at all. If they were sent what method was used? 

6.    What was the sensitivity report if culture were sent?

7.    Without the culture report who was the diagnosis of SSI made?

8.    When did the survelance end?

9.    How were the operated patients followed-up after discharge?

10.What was the size of surgical ward? Number of beds?

11.How operated on the patients during the study period?

Results

12.What was the mean duration of stay?

13.Was the mean duration of stay different in patients with or without SSI?

14.What was the number of patients admitted during the study period?

15.Were all patients that were admitted included in the present study?

16.Presentation of all the tables needs to be improved.

17.It is diffult to follow Figure 1. Please put percentages with bars.

18.Please round off the percentages.

19.Same comment for Figure 2. 

20.The authors have used incidence rate for the SSI rate while it is prevelance 

21.The authors have reported unadjusted odds ratios, while adjusted odds ratios after running logistic regression is most appropriate.

Author Response

Revision:

1.    Rewrite first few lines of introduction keeping only the CDC definition.

                        The paragraph has been recast leaving only the CDC definition. Pg. 2;                   lines 28-33

2.    Page 2 line 49-50 are factually incorrect, please rewrite. E.g. of surgery related factors are-duration of surgery, experience of surgeon, hypoxia during surgery, surgical prophylaxis.

The factors listed are patients-related factors and not surgery related. We want to maintain that the factors listed are correct and are patient related factors. Relevant references have been cited. Pls see Owens and Stoessel. Surgical site infections: epidemiology, microbiology and prevention. J Hosp Infect 2008;70: PAGE 5

3.    Methods-The “surgical prophylaxis” of 5 days cannot be called prophylaxis according to CDC definition

It was a mistake. It has been corrected to reflect both prophylactic and post-operative antibiotics. Pg. 5; lines 106-107

4.    What was the definition of SSI used for the study?

The definition of SSI used is the CDC’s and this is stated clearly in the methodology

5.    What criteria were met to define SSI?

The criteria used for the diagnosis of SSI are clearly spelt out in the CDC document cited and so we felt there was no need listing out the criteria in the manuscript.

6.    It is unclear if cultures were sent at all. If they were sent what method was used?

The focus of this study is on the incidence and factors associated with SSI. According to the CDC document, either of clinical and laboratory findings is an acceptable criterion in the diagnosis of SSI. i.e the presence of either clinical signs or laboratory findings or the diagnosis by the attending physician or surgeon. All these were appropriately communicated to all the surgeons involved in the study and bore in mind while making decision on the status of the wound.

7.    What was the sensitivity report if culture were sent?

Kindly refer to the comment in the query 5 above.

8.    Without the culture report who was the diagnosis of SSI made?

The diagnosis was made by the attending surgeon as stated in the manuscript. Pg. 5; lines 109-110

9.    When did the surveillance end?

The surveillance period has been clearly stated in the manuscript. Pls see page 5, line 97-98.

10. How were the operated patients followed-up after discharge?

This was stated as an obvious limitation of this study. We therefore recommended extended study and development of post-discharge surveillance technique. This was also stated as one of the exclusion criteria.        PG. 14, LINES 254-255

11. How operated on the patients during the study period?

The patients were operated on by their respective surgical team. A consultant surgeon who is also one of the authors of this manuscript and who led a team of surgeons during ward round in which most of the cases were detected was one of the surgeons that operated the patients.        

12. What was the size of surgical ward? Number of beds?

The surgical wards are 32 bed space capacity each. This has been included under the description of study Centre.           Pg.4; line 74-76

13. How operated on the patients during the study period?

This has been addressed in query 10 above     

14. What was the mean duration of stay?  

As requested, the mean duration of stay for patients with SSI and patients with no SSI has been stated in the manuscript alongside the result of corresponding t-test.      Pg. 9, lines 158-160

15. Was the mean duration of stay different in patients with or without SSI?

Yes

16. What was the number of patients admitted during the study period?

135 patients were admitted but 127 patients met our inclusion criteria and were observed. This is stated on page 7, line 134.

15. Were all patients that were admitted included in the present study?

NO. Kindly refer to the response to query 14 above.

17. Presentation of all the tables needs to be improved.

This has been looked into and corrected. We are grateful for this observation.

18. It is difficult to follow Figure 1. Please put percentages with bars.

This has been looked into and corrected. Thanks.

18. Please round off the percentages.

All percentages have been rounded off to one decimal place.

19. Same comment for Figure 2.

This has been looked into and corrected. Thanks.       

20. The authors have used incidence rate for the SSI rate while it is prevalence

Thanks for this important observation. This is a salient issue that was overlooked. After careful reconsideration, we realized that the design of the study was wrongly stated. The appropriate design was actually prospective cohort involving observation of study cohorts over a period of time and comparing the rate of occurrence of SSI among patient with certain characteristics e.g. diabetes and patients without the characteristics. The report was for newly diagnosed cases of SSI. Thus, the rate is an incidence rate.            

21. The authors have reported unadjusted odds ratios, while adjusted odds ratios

after running logistic regression is most appropriate.

This has been corrected. Thanks for this important observation.       

Reviewer 2 Report

The article was written in accordance with the accepted criteria for the journal. I have only a small remark which concerns too modest (poor?) description of the material and methods in the summary.

Author Response

Review Report 1:

Revision:

The article was written in accordance with the accepted criteria for the journal. I have only a small remark which concerns too modest (poor?) description of the material and methods in the summary.

It has been elaborated to give a more detailed description of the methods in the abstract including an insight into the statistical analysis. Pg. 1; Lines 8-14

Reviewer 3 Report

The study is aimed at investigating the rate and determinants of SSI at a tertiary hospital in Nigeria.

Although the topi of SSI in Africa/LMICs is of interest, the originality of the submitted manuscript is low and the quality is poor.

It could be considered for publication after being extensively reviewed and resubmitted as short report.

Please find below some of the mayor revisions suggested.

Introduction

The Introduction should be extensively reviewed and shortened. For example, at the beginning of the paragraph, there is no need to specify all the definitions of SSI. I suggest to focus the background more on SSI in LMICs, highlighting the lack of information and proper surveillance.

Material and Methods

1. Study design: why did the Authors decide to use a cross sectional study and not a prospective cohort study? May they spend one sentence in the Methods about that?

2. Move Ethical Consideration at the end of the paragraph.

3. Why gynecological and obstetric patients were excluded form the study? CS definitely represents the most performed surgery in LMIC (see also De Nardo P et al. J Hosp Infect, 2016) .

4. I suggest to delete the sub-heading "monitoring patients for developing of SSI" because the Author used it also to explain the patients' enrollment.

5. "Prophylactic empiric antibiotic covering, for 5-7 days" is not acceptable. It should be rephrased "post-operative antibiotic course".

6. SSI categories: please use superficial, deep and involving organs and spaces as per CDC classification.

Results

Risk factors are not well investigated/explained (skin preparation, pre-operative prophylaxis YES/NO, surgical technique, suture, etc.)

No need of table 1.

Comments

1. The Authors have already published a manuscript about SSI at the same facility. They should clearly state which is the reason to repeat the same study after very few years.

2. Not all the conclusions are supported by the results (i.e.: no investigations about microorganism causing infections have been performed).

3. The Authors should better speculate about importance of surveillance in LMIC (see, among the others, Nguhuni B et al. Antimicrob Resist Infect Control, 2017).

Author Response

Review Report 3:

1     The Introduction should be extensively reviewed and shortened. For example, at the beginning of the paragraph, there is no need to specify all the definitions of SSI. I suggest to focus the background more on SSI in LMICs, highlighting the lack of information and proper surveillance.

Following the kind advice of the reviewer, the introduction has been reviewed with special focus on the SSI in LMICs.           Pg. 2; lines 29-33 and pg. 3; lines 52-61.

2     Study design: why did the Authors decide to use a cross sectional study and not a prospective cohort study? May they spend one sentence in the Methods about that?

Critically looking at the study, we realized we wrongly stated the design as cross sectional when it was prospective cohort study involving observation (following up) of study cohorts over a period of time and comparing the rate of occurrence of SSI among patient with certain characteristics eg diabetes and patients without the characteristics. Pg 4; line 89

3     Move Ethical Consideration at the end of the paragraph.

This has been done.            Pg. 6 line 127-134

4     Why gynaecological and obstetric patients were excluded from the study? CS definitely represents the most performed surgery in LMIC (see also De Nardo P et al. J Hosp Infect, 2016)

The hospital has four surgical wards-male, female, paediatric and gyneacological wards. Based on the available resources (human and material), we decided to concentrate on the general surgical wards why recommending an extensive study to cover all service units of the hospital

5     I suggest to delete the sub-heading "monitoring patients for developing of SSI" because the Author used it also to explain the patients' enrolment.

Noted sir. Thanks for the observation. The two sections have been merged together.            

6     "Prophylactic empiric antibiotic covering, for 5-7 days" is not acceptable. It should be rephrased "post-operative antibiotic course".

It was a mistake. It has been corrected to reflect both prophylactic and post-operative antibiotics.     Pg. 5; lines 106-107

7     SSI categories: please use superficial, deep and involving organs and spaces as per CDC classification.

                        This was actually the categories used as this study was guided by the CDC                       document      on the guidelines for the prevention of SSI.        Pg. 6; lines 115-                   116

8    Risk factors are not well investigated/explained (skin preparation, pre-operative prophylaxis YES/NO, surgical technique, suture, etc.)

We only investigated routinely reported factors in our environment. All patients received pre-operative prophylaxis and post-operative antibiotics. In most cases, the same set of antibiotics are used as stated in the manuscript.

9.   The Authors have already published a manuscript about SSI at the same facility. They should clearly state which is the reason to repeat the same study after very few years.

The previous study was retrospective in design and reported the prevalence of SSI in the hospital between the year 2013 and 2014. The present study is a prospective study conducted in the same facility and reporting the incidence of SSI within the specified study period.

10. Not all the conclusions are supported by the results (i.e.: no investigations about microorganism causing infections have been performed).

As stated in our objective, we sought to investigate the occurrence and factors influencing SSI in our facility. We believe our conclusion is in line with our stated objective.

11.            The Authors should better speculate about importance of surveillance in LMIC (see, among the others, Nguhuni B et al. Antimicrob Resist Infect Control, 2017).

Thanks for the recommendation. This has been taken into consideration.

Round 2

Reviewer 1 Report

Write short forms used at the bottom of the table 2. 

Most of my other concerns have been addressed.